# How Distressed Are Adolescent Students? A Mix-Method Study on High School Students in Northern Italy, Two Years after the Beginning of the COVID-19 Pandemic

**DOI:** 10.3390/bs14090775

**Published:** 2024-09-03

**Authors:** Anna Maria Della Vedova, Loredana Covolo, Carlotta Fiammenghi, Silvia Marconi, Umberto Gelatti, Maurizio Castellano, Barbara Zanini

**Affiliations:** 1Department of Clinical and Experimental Sciences, University of Brescia, 25123 Brescia, Italy; silvia.marconi@unibs.it (S.M.); maurizio.castellano@unibs.it (M.C.); barbara.zanini@unibs.it (B.Z.); 2Department of Medical and Surgical Specialties, Radiological Sciences and Public Health, University of Brescia, 25123 Brescia, Italy; loredana.covolo@unibs.it (L.C.); carlotta.fiammenghi@unibs.it (C.F.); umberto.gelatti@unibs.it (U.G.)

**Keywords:** study engagement, adolescents, school, youths, stress, eating behaviors, smartphone overuse

## Abstract

Adolescence is a central phase for the development of a person’s identity, involving complex multidimensional changes and increasing vulnerability to distress. This study aimed to investigate the psychological well-being of adolescent students in Brescia (Northern Italy), two years after the beginning of the COVID-19 pandemic. An online survey investigated the extent and the reasons reported by high school students (13–21 years) for their emotional distress (ED), also considering other factors such as physical activity, nutrition, sleep and smartphone overuse. The main reasons for ED were classified through a qualitative analysis of the free-text answers. A total of 1686 students agreed to participate, and 50% showed a presence of ED. According to a multiple logistic analysis, adolescents were more at risk of ED if they were female (AdjOR 2.3), older (AdjOR 1.6), slept less than 8 h (AdjOR 1.5), perceived increased anxiety (AdjOR 3.4), and adopted certain eating behaviors, e.g., comfort food consumption (AdjOR 2.0). According to free-text answers, the main reasons for ED were “school pressure”, “relationships with family and peers”, “negative emotional states” and “the pandemic”. The results show a high level of ED among adolescents, and the reported reasons may help to better address their psychological needs after the pandemic.

## 1. Introduction

The COVID-19 pandemic significantly affected the daily life of millions of people around the world, and Italy was the first, among Western countries, to introduce strict social restrictions [1,2,3]. Initially, lockdown measures were limited to the Lombardy region, in Northern Italy, and few days later the restrictions became national [4,5]. During the following two years, Brescia and Bergamo districts, both in Lombardy, remained the two cities with the highest alert level and with the strictest isolation measures. It is well-known that social relationships are an important determinant of health, alongside education, lifestyle, environment, employment and working conditions [6]. Physical distancing, online schooling, restrictions on social meetings and physical and outdoor activities profoundly affected children’s and adolescents’ habits. An increase in psychological distress and a high prevalence of COVID-19-related fear were reported among children and adolescents from the beginning of the pandemic [7,8].

Adolescence is commonly considered as the developmental phase that begins with the physiological changes of puberty and ends with the establishment of a personal identity and adult roles. According to the World Health Organization (WHO), adolescence roughly corresponds to the period between 10 and 19 years. Recently, the age range has been broadened due to the prolonged process of achievement of adult roles which is common in contemporary societies [9]. The American Academy of Pediatrics suggests 11–21 years as an optimal interval that includes the different stages of adolescence [10].

Adolescence is a formative period in life that encompasses several physiological and psychological changes which directly affect lifelong health. Adolescents go through several stages of great complexity, given that this period of development is pivotal for psychological and biological changes and can be decisive in setting individual behavioral trajectories which influence adulthood [3,6,7,8,11].

In the long trajectory that leads from childhood to adulthood, young people are faced with a range of developmental tasks, such as mastering skills, forming close relationships, exploring different roles, accepting bodily changes, establishing gender identity, envisioning their future, and acquiring the essential abilities and values needed for a successful transition into adulthood. Following Erickson’s (1968) theory of psychosocial development, the main developmental task of adolescence is the adult identity formation, namely the awareness a person has of being him/herself and different from others, including all personal attributes and the feeling of continuity over time, despite changes [12]. Furthermore, to define his/her own identity, the adolescent must renegotiate aspects of separation from parental figures and build aspects of him/herself on which to base his/her own personal identity. These processes imply great emotional intensity and their resolution is not obvious, but is the result of strenuous psychic work which Erikson (1968) defined as a “maturational crisis” [12].

Notably, the concept of “crisis” is positive and dynamic in Erikson; nevertheless, the complexity of fulfilling these developmental tasks makes adolescence a phase of increased psychological vulnerability. Recent literature highlights the fact that mental health needs in young people had grown over the past 10 years before the pandemic. According to the WHO (2021), 13% of 10–19-year-olds experience mental health issues which are largely unrecognized and untreated [13]. A study prior to the pandemic estimated the global prevalence of unmet needs of mental health care among adolescents as being more than 50% [14].

In this scenario, the COVID-19 pandemic intervened as a further factor negatively affecting the emotional well-being of the adolescent population. In particular, the confinement associated with the pandemic interfered with their daily lives, affecting adolescents’ vital need for social interaction with peers, disrupting the normal rhythms of school life, hindering the possibility of outdoor leisure activities and of carrying out regular sporting activities. In this context, the perception of loneliness has increased among adolescents, which could have played a part in the psychological suffering of young people during the pandemic. An extensive literature has underlined the critical impact of the COVID-19 pandemic on the rates of anxious, depressive and psychopathological disorders in the general population and especially in young people [15]. A meta-analysis study on the global prevalence of anxiety and depression in children and adolescents found a likely doubling of psychological difficulties since the start of the pandemic [16]. The pooled prevalence of clinically elevated symptoms of depression and anxiety in the pandemic period reached 25.2% and 20.5%, respectively, while before the pandemic prevalence was at 12.9% for depression and 11.9% for anxiety. Anxiety and depression symptoms showed an upward trend as the pandemic progressed, and were higher in girls, while depressive symptoms increased with age [16].

These results are confirmed in a further study in the USA, where there was a 12% increase in the odds of adolescents using Mental Health Services during COVID-19 and the odds ratio of females compared to males increased significantly, by 63%, during the pandemic [17]. A recent study carried out in Italy over the different periods of the pandemic (from March 2020 to May 2022), pointed out that the access to emergency departments for acute psychiatric issues, both those for any reason and those for suicide attempts in children and adolescents, grew impressively (a two-fold increase) from the period before the pandemic to its conclusion, peaking in the second wave [18].

Quality of life, defined as a broad concept typically referring to an individual’s overall well-being, encompassing physical health, psychological state, level of independence, social relationships, personal beliefs, and their relationship to salient features of the environment, was also severely affected during the pandemic [19]. Several authors explored changes in physical activity, mental health, sleep quality, screen use and eating behavior [20,21]. Xiang et collaborators detected a worrying increase in sedentary behaviors and time spent in front of screens among adolescents in China, with the percentage of physically inactive students increasing from 21.3% to 65.6% [22]. Bruni et al. reported changes in sleep habits with difficulties in falling asleep and an increase in screen use in Italian adolescents [23]. A survey on children and adolescents living in South Tyrol, in the North of Italy, conducted in 2021, confirmed the presence of mental health problems, emotional symptoms and poorer quality of life after the first year of the pandemic, especially in children and adolescents with low socioeconomic status, a migration background, and limited living space [24]. These findings were aligned with a similar study conducted in Germany where two-thirds of the children and adolescents reported being highly burdened by the COVID-19 pandemic, experiencing a reduction in the quality of life and an increase in anxiety and mental health problems compared to the pre-pandemic period [25]. In the large Norwegian COVID-19 Young study, comprising participants aged 11–19 years, Lehmann and co-workers noted a significant increase in internalizing problems between the lockdown in April/May 2020 and 9 months later, in December 2020/January 2021 [26]. In a recent systematic review, Orban and collaborators concluded that symptoms of depression, anxiety and mental health issues in children and adolescents increased during the pandemic compared to before, and remained high for a long period after the onset of the pandemic [27].

Not many studies have looked at changes in eating behaviors in young people during the pandemic [28]. Zanini and colleagues found a general improvement in the eating habits of adolescents in northern Italy [29]. Nonetheless, another Italian study found an increase in eating disorders in male and female adolescents during the pandemic [30].

To assess the extent of long-term changes in emotional well-being and several aspects of lifestyle, including eating habits, physical activity and technology addictions, we designed a web-based survey addressed to adolescents, two years after the onset of the COVID-19 pandemic, during a period of slow “return to normal”.

Considering the amount of literature documenting the increase in emotional distress among young people during and following the pandemic, exploring the reasons for adolescents’ discomfort in more depth may provide worthwhile insight to promote better psychological adjustment and developmental outcomes. To this aim, we dedicated part of the research to collecting the reasons for distress through free-text questions, to capture its nuances and contents more richly. The presence of emotional distress in adolescents during the pandemic has been documented in numerous studies as an increase in psychological symptoms including anxiety, sadness, depression, loneliness, restlessness and fear [31]. Specific vulnerabilities of adolescence are also the fear of failure, self-doubt and loss of motivation [12], themes that we expect to emerge in free-text answers.

Investigating these aspects among adolescents may also contribute to addressing public health actions to prevent the development of numerous chronic or degenerative diseases in adulthood [32].

Following the recently published findings on changes related to eating habits during the pandemic, in this article we analyzed the effects of the pandemic on the psychological well-being of the adolescent population in the province of Brescia, even when strict restrictions or lockdown measures were no longer in place [29,33]. Considering that such a large study on the Italian adolescent population has not been carried out before, in line with previous research carried out in other geographical contexts, we hypothesized that, also in our sample, emotional distress would increase compared to the pre-pandemic period. We also hypothesized that ED would be associated with different aspects of lifestyle such as eating habits, difficulty sleeping, and overuse of smartphones and digital devices. Finally, we hypothesized that ED would be associated with the presence of difficulties related to school commitments and to relationships with peers and with relatives, possibly exacerbated by the restrictions and changes imposed by the pandemic.

## 2. Materials and Methods

The results of the present study are a part of a larger cross-sectional observational study, named COALESCENT (Change amOng ItAlian adoLESCENTs) and conducted in collaboration with and under the supervision of the Territorial School Service of Brescia District. The main tool of COALESCENT was a web-based survey developed using the open-source software LimeSurvey 6.6.1 (LimeSurvey GmbH, Hamburg, Germany). We submitted the online survey proposal to all high schools of the Brescia district (39 state and 18 state-authorized private schools); among them, eight schools agreed to collaborate. Students attending these schools and willing to participate provided an online informed consent before starting the questionnaire, without any other selection criteria; for underaged students, a parental consent (following approved procedures in each school) was provided; the survey was administered during school hours and teachers were present and available to provide explanations, upon request. According to Italian law, this study did not require approval by the Ethics Committee, because it was completely anonymous since the participants could not be traced or identified once the questionnaire had been sent [34]. The questionnaire consisted of 110 questions, divided into 8 sections. For the purpose of the present sub-analysis, in this section we provide details about sections F (questions about physical and sport activity, duration and quality of sleep, duration and type of screen activity, smartphone addiction) and G (6 questions about emotional distress). Further information about methodological, technical and ethical details of the study are provided in previous publications [29,33].

Section F. Physical activity was investigated with six questions, then grouped into a single data set indicative of a high, medium or low level of physical activity performed daily by the participants. Two questions explored if they were engaged in a sports club before the pandemic period and if they definitely dropped out after the period of COVID restrictions. Three questions collected self-reported sleep duration and changes in sleep quality and duration, compared to the pre-pandemic period. Time spent in screen activity, in comparison to the period before the pandemic, was self-reported and divided into three categories: for study, for fun and out of boredom. To assess smartphone addiction, we used the validated questionnaire, Smartphone Addiction Scale-Short Version (SAS-SV; [20,35]). This scale is a 10-item questionnaire developed in Korea measuring the risk level of smartphone use with the aim of identifying high-risk groups among adolescents. The total score ranges from 10 to 60, with higher scores reflecting higher problematic use of a smartphone in the past year. The Italian version was validated by maintaining the original cut-off score indicating probable smartphone addiction (31 for males and 33 for females) [35]; a further Italian study suggests a two-factor structure, namely “addiction” (which refers to the emotional involvement and irresistibility of attraction) and “functional impairment” (which refers to the degree of interference in carrying out daily activities and studying) [36].

Section G. The tool investigating emotional distress was the Matthey Generic Mood Questionnaire (MGMQ), a standardized short questionnaire assessing the presence, impact and reasons for possible emotional distress [37,38]. The MGMQ consists of 4 questions, which we have slightly adapted for this study compared to the original: the first (Distress) question asked, “In the last 2 weeks, have you felt any of the following for some of that time: very stressed, anxious or unhappy, or found it difficult to cope?” Response options were “yes”, “possibly” or “no”. Respondents who answered “yes” were asked the second (Bother Impact) question, “How bothered have you been by these feelings?” Response options were “not at all”, “a little bit”, “moderately” or “a lot”. The threshold for having a potentially clinically relevant distress condition is having chosen the “a lot” or “moderately” bothered response option. Based on these responses, a binary variable “distress” with option “yes/no” was created, where “no” included adolescents who answered “possibly” or “no” to the first MGMQ question in addition to adolescents who answered “not at all” or “a little bit” to the second MGMQ question. A third “Reason for Distress” question (if the Bother Impact question was endorsed) is an open question aimed at assessing the reasons behind the distress. A fourth question ‘Wish for Referral’ was modified by asking participants if in the last two years they had turned to professionals such as psychologists or psychotherapists more often compared to the pre-pandemic period. The MGMQ has been validated in the field of perinatal emotional well-being and has been translated and adapted to the Italian population [38,39].

### 2.1. Statistical Analysis

The analyses included descriptive statistics (i.e., frequencies and percentages for categorical variables and mean values with standard deviations for continuous variables). Comparisons between groups were made using the chi-square test for categorical variables and the Mann–Whitney test for continuous variables. A binary logistic regression model was carried out, with “distress” as the dependent variable. The covariates to be included in the final model were selected on the basis of univariate analysis, with a univariate *p* value < 0.05 as the main criterion. Then, using a backward selection process, statistically non-significant variables were excluded. To check for collinearity among the variables, the Spearman correlation test was used. Internal consistency was measured using Cronbach’s alpha with a range between 0.70 and 0.95 considered acceptable [40]. The results of the logistic regression are reported with adjusted odds ratios and 95% confidence intervals. A *p*-value less than 0.05 was considered statistically significant for all analyses. Statistical analyses were performed using STATA (Stata Statistical Software: Release 18.0, College Station, TX, USA: Stata Corporation).

### 2.2. Analysis of Free-Text Answers

Free-text answers were evaluated through both a lexical and a thematic analysis. The lexical analysis was carried out by copying the answers in a txt file which was uploaded to the software MaxQDA (Verbi Software 2024). The software was used to calculate the frequencies of each word used in the corpus and produced a word-cloud, that is, an image made of words, whose size varies according to the frequency with which they recur. Word clouds are constructed from the most frequent content words (i.e., nouns, adjectives, verbs, and adverbs) in the corpus; an exclusion list was uploaded to the software, so as to exclude function words (i.e., determiners, prepositions, auxiliaries). These are usually excluded from linguistic analyses based on frequency, because they are, by their very nature, very frequent linguistic elements in a given language and therefore not characteristic of the individual corpus of texts examined; conversely, content words possess semantic content and contribute to the meaning of the text in which they are used. The word cloud includes word forms instead of lemmas, i.e., each possible form of a word as it is used in the corpus instead of one single umbrella term comprising all its possible word forms. This is useful for identifying the differences between, for example, the use of the singular noun “situation” (as in “the general situation makes me feel anxious”) and the use of the plural noun “situations” (as in “different situations making me feel anxious”), as well as noticing patterns of usage (for example, verbs used in the first-person singular form). The complete list of words included in the word cloud as the most frequent words in the corpus, together with their overall frequencies, is given in the Appendix A. Word forms belonging to the same lemma are grouped together in the table.

The thematic analysis was carried out through close reading of each answer. The authors conducted an inductive semantic description of the data set following the guidelines provided by Braun and Clarke [41]. This meant that two researchers thoroughly familiarized themselves with the contents of each answer through careful reading and re-reading; the researchers then coded the answers by identifying each theme expressed in each text. In many cases, participants expressed more than one cause for distress: in these cases, the answers were coded multiple times in order to record each cause for distress. This provided a more accurate description of the contents of each answer, and also limited the influence of the researchers’ own sensitivity to particular themes, or of the researchers’ own experiences with ED, on the interpretation of the texts.

## 3. Results

The online survey was completed by 1686 students (response rate 34.6%) attending eight different high schools within the Brescia district in the North of Italy; 77% of students were recruited from state-funded schools. Females accounted for slightly more than half of the sample (n = 846; 52%); 3.2% (n = 54) of the participants chose not to declare their gender. Median age of the sample was 16 years, ranging from 13 to 21 years. About 60% of students declared having at least one parent with a university degree. Using the SAS-SV cut-off, smartphone overuse was found in 46% of adolescents (n = 772). Possible presence of psychological distress was found in half of the sample (n = 848).

### 3.1. Predictors of Psychological Distress

The predictors of psychological distress were investigated by performing both univariate and multivariate analyses, as shown in Table 1.

The multivariate logistic analysis showed that adolescents aged 16–18 years old (AdjOR = 1.64, 95% CI 1.29–2.08) and females (AdjOR = 2.27, 95% CI 1.77–2.90) were more at risk of having psychological distress. This condition was also associated with eating less (AdjOR = 1.90, 95% CI 1.35–2.69) and eating less healthily (AdjOR = 1.63, 95% CI 1.07–2.49) compared to the pre-pandemic period. The consumption of comfort food was associated with psychological distress, regardless of changes occurring compared to the pre-pandemic period (AdjOR = 2.02, 95% CI 1.44–2.83). Adolescents who declared to sleep less than 8 h (AdjOR = 1.45, 95% CI 1.14–1.85) were also at increased risk of having psychological distress. As regards smartphone addiction, every additional point increased significantly the risk of having psychological distress by 2% (AdjOR = 1.02, 95% CI 1.00–1.03). Taking into consideration the two-factor structure of SAS-SV instead of the one-dimensional one, the variable associated with psychological distress was the “functional impairment” factor (AdjOR = 1.08, 1.04–1.11; *p* < 0.0001) rather than the “addiction” factor (AdjOR = 0.97, 0.95–1.0; *p* = 0.1). The Cronbach’s α of SAS-SV was 0.82, and those of “functional impairment” and “addiction” factors were 0.69 and 0.82, respectively.

Finally, increased anxiety perception (AdjOR = 3.37, 95% CI 2.88–4.68) and need of health professional support (AdjOR = 1.84, 95% CI 1.29–2.62) compared to the pre-pandemic period were both associated with psychological distress.

### 3.2. Time Spent on Device According to Possible Presence of Psychological Distress

Compared to the pre-pandemic period, time spent on digital devices increased a lot, especially for study (54%) compared to leisure (28%), or out of boredom (30%). As shown in Figure 1, psychological distress was significantly higher among students who declared to have increased time spent on digital devices for study (61% vs. 48%) (a) and out of boredom (36% vs. 23%) (c) (*p* < 0.0001), as opposed to students who declared unchanged time. No relation with psychological distress was found in the case of time spent on digital devices for leisure (c).

Students were asked to report approximately the hours spent on digital devices. Data confirmed previous results, even though we did not obtain answers from each participant. Students with possible psychological distress, particularly, (n = 786) declared spending 4.3 ± 2.8 h using digital devices for study compared to 3.7 ± 2.6 h reported by students without psychological distress (n = 716) (*p* < 0.0001). Similarly, time spent using digital devices due to boredom was significantly higher among students with possible psychological distress (n = 725) compared to others (n = 681) (3.2 ± 3.2 vs. 2.3 ± 2.3; *p* < 0.000, respectively). No significant difference was found in the case of time spent on digital devices due to leisure in the two groups (3.7 ± 2.9 vs. 3.8 ± 3.2, *p* = 0.8).

### 3.3. Free-Text Answers

Among the participants who said that they were experiencing a state of distress, 770 (77%) decided to type in an answer further specifying the causes for their distress. Of these, 20 answers were excluded from the analysis because they were evidently nonsensical or unintelligible. Considering the valid answers, we calculated that 58% of the participants who stated that they were “a little” bothered by their feelings of distress decided to elaborate with a written text; percentages were as high as 70% for those who stated they felt “quite” bothered, and 76% for those who answered that they felt “very” bothered. Moreover, we calculated that 43% of the adolescents aged 13–15 years old, 47% of adolescents aged 16–18 years old, and 43% of adolescents aged 19 years old or older decided to type in an answer (these categories correspond roughly to the different stages of adolescence as identified by the American Academy of Pediatrics [10]).

#### 3.3.1. Lexical Analysis

The texts varied greatly in length (max.: 1767 words, min.: 1). The quantitative analysis of the lexis used in these answers revealed that the most frequently used content words belonged to the semantic spheres of school (i.e., “scuola/school”, 309 hits; “studio/study”, 87 hits; “verifiche/written tests”, 58 hits; “interrogazioni/oral tests”, 38 hits;, “compiti/homework”, 16 hits; “professori/teachers”, 15 hits; “studiare/to study”, 14 hits;); of interpersonal relationships (i.e., “amici/friends”, 57 hits; “persone/people”, 45 hits; “famiglia/family”, 39 hits; “genitori/parents”, 27 hits; “relazioni/relationships”, 12 hits); content words related to negative emotional states (i.e., “ansia/anxiety”, 50 hits; “stress”, 48 hits; “problem/problems”, 40 hits; “pressione/pressure”, 31 hits; “paura/fear”, 19 hits; “difficoltà/difficulties”, 14 hits); and words related to the pandemic (i.e., “pandemia/pandemic”, 31 hits; “COVID”, 23 hits). These results are shown in the word cloud in Figure 2. As previously mentioned, the complete list of the words included in the word cloud, together with their translation into English and their frequencies in the free-text answers, is given in the Appendix A.

#### 3.3.2. Thematic Analysis

As explained in more detail in the Methods section, the content analysis aimed to identify the causes of distress mentioned by the adolescents in their free-text answers. Therefore, each cause for distress was codified, and the answers where adolescents mentioned more than one cause for distress were coded multiple times. Overall, 509 participants were able to clearly identify only one major cause for distress, while 241 participants mentioned more than one. Seven topics were thus identified as major causes for ED; these topics largely overlap with the semantic categories of the most frequent words in the corpus, highlighted through the lexical analysis:School: school, study load, and distress caused by teachers, tests and marks were mentioned in 428 answers (57%).Interpersonal relationships: difficulties relating to classmates, friends, and/or family members were mentioned in 218 answers (29%).Negative emotional states: negative emotional states such as anxiety, fear of failure, or lack of motivation were mentioned in 107 answers (14%). Some of these answers (12% of the subset, 1.7% of the total) described profound psychological distress such as depression, eating disorders, panic attacks, self-harm, and suicidal ideation.COVID-19 pandemic: 85 answers (11%) mentioned the consequences of the restrictions and/or fears of catching COVID-19.Overall negative, but otherwise unspecified, emotional state: 68 answers (9%) described a general, but otherwise unspecified, feeling of distress, and/or the writer’s inability to identify one or more specific cause/s for distress.Feelings of isolation and loneliness: were mentioned in 43 answers (6%).Other: this category comprises 71 answers (9%) where participants mentioned other, less frequent causes for distress, such as concerns for one’s physical appearance and/or food behaviors, mentioned in 22 answers (3%); distress caused by sports activities and training, mentioned in 16 answers (2%); distress caused by overuse of digital devices and/or social media, mentioned in 13 answers (2%); tiredness and/or poor sleep quality, mentioned in 9 answers (1%); dissatisfaction with one or more personality traits, mentioned in 5 answers (0.7%); and dissatisfaction with one’s job and/or economic situation, mentioned in 4 answers (0.5%).

A total of 14 answers (2%) were written by participants who stated that they did not want to specify the reasons for their feeling of distress.

The presence of distress was significantly higher only for those who reported relational reasons (90% vs. 78%, *p* < 0.0001) and negative emotional states (90% vs. 80%, *p* = 0.016). Considering negative emotional states, older people suffered more (16.2 y vs. 15.8 y, *p* = 003), even though the significance was weak.

In the following subsection, a selection of examples of answers given by participants for each category is provided and described. Examples are here translated verbatim from Italian into English; the original Italian texts can be found in the Appendix A.

#### 3.3.3. Examples from the Corpus

Clear examples of answers falling into the first category (school) are 166 texts in which participants simply wrote “school” or “heavy study load”. Other participants decided to elaborate more on the topic, often mentioning the pressure exerted on them by relatives and/or teachers, a sense of overwhelming and the fear of disappointing parents’ expectations, such as:School makes me feel anxious. I feel as if I couldn’t face all the pressure that is exerted on me by all the things I have to do and the expectations. (Girl, 14 years old)I think the main causes have been: school activities and the pressure exerted on me by my parents due to school. (Girl, 15 years old)Accumulation of written/oral tests because of the end of the term. I felt the pressure of doing well at school and having not to disappoint expectations. (Girl, 18 years old)

A smaller, but non-negligible number of respondents (N = 16) report that their teachers lack understanding and support, as in the following:The overload of homework and the teachers’ lack of understanding of the fact that we are coming out of a difficult period. They increase homework and punish more often, they are more frustrated than before!!! (Boy, 15 years old)School has recently made me feel very anxious because of tests and some teachers who don’t care about their students. (Boy, 15 years old)

Ten participants were dissatisfied with distance learning and teaching but a similar number of respondents (N = 7) felt uncomfortable returning to in-presence school after COVID-19:Because of covid, distance learning, studying increased considerably. So I was tired at the end of the day because I was studying too much and had too much homework. I no longer had time for myself, to have fun, etc. … I used to spend hours and hours in front of the computer every day at all times. (Girl, 17 years old)The return to in-presence school has upset me a lot, I cannot keep up with the lessons, focus and study. (Girl, 17 years old)Impressively, some respondents displayed detrimental consequences of their huge distress related to school:I think school has made me feel depressed and changed me for the worse because I don’t feel inspired to do anything anymore. (Boy, 14 years old)Too much study, the overlapping written and oral tests. My mind was shattered and my body felt the consequences. (Girl, 15 years old)I had problems at school: I suffer from panic attacks and anxiety and when I have too many things to do I get anxious. I have attempted suicide twice in twenty days taking an exaggerated number of pills, even ending up in hospital. I felt overburdened with school and full of responsibilities, also due to the fact that I am a senior in high school. (Girl, 18 years old)

The second category of answers related to adolescents’ relational problems with their peer group (friends, classmates, romantic partners) and with family members, both associated with painful feelings, as in the following:Arguments with my classmates, they have “insulted”, excluded, isolated me, and they have turned most of the class against me. (Boy, 14 years old)Not feeling understood, being less energetic, feeling a void inside due to disappointments caused by people who were once important in my life and consequently not being able to trust people anymore. (Girl, 15 years old)The causes are my oppressive parents who would not accept my homosexuality, given that they already do not accept me. (Girl, 13 years old)They keep telling me the same things over and over, it’s tiring. My parents should talk about something else besides school, otherwise my self-esteem keeps getting lower and lower. (Boy, 15 years old)The lack of relationships has caused a loss in my ability to relate with others, causing a feeling of inadequacy when I am in a group. (Girl, 15 years old)

The third category included answers mentioning negative emotional states, mostly “anxiety”, which is mentioned 54 times in the corpus, as in the following:There are many things that make me feel anxious, I am constantly feeling strained. (Girl, 18 years old)

Some worrying answers included in this set also described profound psychological distress leading to self-harm, sometimes linked to other factors such as school and the pandemic, as in the following:Depression and anxiety which have caused difficult situations such as self-harm. (Girl, 15 years old)Lately I have been losing the desire to go out, I prefer to stay in my comfort zone, in my room. I no longer find stimuli to go out and I am not interested in taking care of myself. […] I have also noticed that I have been more sensitive and irritable since the pandemic started. In the initial period of this tragedy I unfortunately started to feel very lonely and I found relief in self-harm, which I have, sadly, recently resumed. (Girl, 13 years old)My mind during this long period coinciding with the beginning of the pandemic until today has undergone a radical change, and today the situation is certainly very negative compared to before, due to the emergence of countless situations of discouragement, panic attacks and moments of pressure or depression, which have had a great emotional impact. Certainly all this is also linked with the emergence of problems and obsessive behavior towards my body, which has totally changed due to a loss of more than 30 kg, both in terms of training and above all nutrition, which today have led me to have to follow a psychological and nutritional counseling course. (Boy, 17 years old)I started thinking about life, how it could end at any moment. I started to think that I am not enough for others (but it’s me who thinks so). I am also frightened of myself because I sometimes think about the option of suicide and I am terrified just thinking that I might do something that could make other people suffer. (Girl, 15 years old)

The fourth category includes answers which mention the consequences of the pandemic, especially in terms of the negative impact of the restrictions to prevent contagion. Some participants simply wrote “COVID restrictions” or “having to stay at home”, while others described more precisely their suffering at “being separated from their friends/partners/family”, or having to give up sports and hobbies, as in the following:The fact that I stayed at home with my parents and my sister and I have not seen my friends for a long time. (Girl, 15 years old)I think the fact that I stayed at home and never went out, that I no longer had contact with people and did not play the sport I loved. (Boy, 15 years old)

Six participants were afraid of falling ill, and three described their experience of being ill. Discordant voices were uncomfortable with the end of the restrictions (N = 3) and preoccupied with life after COVID-19 (N = 2).

Also in relation to this topic, states of great psychological suffering and self-harming thoughts have emerged:During the pandemic I had a lot of time to think and to spend time with my very toxic family, and this has harmed me a lot, and several times I have thought about ending it, I was and still am mentally shattered but no one notices it. (Girl, 14 years old)

The fifth category included answers where participants felt unable to identify one or more definite causes for their distress. A total of 31 participants clearly wrote “I don’t know”; others wrote answers such as “the general situation”, “my life in general”, or:A general dissatisfaction with several aspects of life which I am struggling with and which make me unable to focus on other aspects which I should be more serene about. (Boy, 18 years old)Well I don’t know… I don’t know, there’s no valid reason. (Girl, 15 years old)Inexplicable anxiety, for I don’t know what! (Girl, 17 years old)

The sixth topic included 43 answers explicitly mentioning a pervasive feeling of isolation and loneliness, which could be related either to interpersonal problems or to the consequences of the pandemic. Typically, participants expressed a “feeling of loneliness”, “feeling of isolation” (N = 14), or wrote “I feel lonely” (N = 6), as in the following:The fact that I felt extremely lonely, I was looking for help but no one could help me, neither within the family nor outside. (Boy, 13 years old)

Finally, less frequent but equally significant topics were grouped under the label “other”, including concerns for one’s physical appearance and food behaviors, distress caused by overuse of digital devices and/or social media, and poor sleep quality, as in the following:Probably this difficult situation we find ourselves in, digital devices which are addictive. This in my case heavily affects sleep, it’s difficult to fall asleep and I often wake up in the middle of the night. (Girl, 14 years old)I waste too much time with my smartphone and I can’t restrict its use, I find it difficult to organize myself with my studies both because of the smartphone and because I struggle to focus, I am aware that I’m wasting my time but I can’t change this situation. (Girl, 15 years old)Problems with my body and food. (Girl, 16 years old)

## 4. Discussion

The objective of the study was to evaluate the presence and reasons for distress among adolescent students from northern Italy in the final phase of the pandemic, together with its impact on their well-being and lifestyles. The vast majority of young participants said they felt definitively (65%) or possibly (10.8%) distressed. Furthermore, the percentage of participants who admitted to experiencing a fairly or very disturbing distress situation (above the threshold of clinical interest) was notable, exceeding 50%. This result places our study in line with the other studies carried out in Italy and elsewhere in the world, finding a worrying increase in the psychological distress of young people during the pandemic and in its subsequent phases [15,16,18].

One of the aims of the study was to analyze the factors most associated with adolescent distress. As in previous studies, in our sample symptoms of distress increased with age and were higher in girls [16,17,18,42,43]. The most distressed age group was between 16 and 18 years old, a moment of middle-to-late adolescence that is central to the development of identity and where relationships with peers are of enormous importance [12]. In adolescence, the crucial role of context and social experiences for brain development and for the construction of identity has increasingly been recognized. During these phases, the progressive maturation of the prefrontal cortical areas elevates reflective thinking skills, the ability to evaluate dangers, self-judgment, self-image, and the ability to assess one’s skills in comparison with peers and with respect to adults’ expectations, making sociality a decisive aspect [44]. Thus, this result seems to underline the fact that middle adolescence is a sensitive phase with respect to the challenges presented by the pandemic, especially social restrictions and the threat posed by the COVID-19 disease. Furthermore, the phase between 16 and 18 years is the one in which young people are most focused on choices related to their academic future. These are influenced by the skills they feel they have acquired, and therefore the sensitivity to school failure is very high in this phase. Therefore, the school pressure linked to the changes due to the pandemic, as highlighted by the answers to the free-text questions, may also be at the root of the increase in ED.

The double percentage of girls suffering from clinical distress is a very worrying fact, already known but exacerbated by the pandemic, which should lead to thinking about prevention projects that pay attention to gender aspects.

Several lifestyle factors were significantly associated with distress, while socio-demographic aspects such as level of parents’ education or the type of school attended were not. One aspect which was specifically investigated by this study were adolescents’ eating habits. A new and interesting result is that adolescents who pointed out that their eating habits had worsened during the pandemic, said that they ate less, and made greater use of comfort food, were also those at greater risk of emotional distress. This result supports the idea that psychological well-being and correct nutrition are two profoundly linked aspects and that nutrition must be taken into account when designing interventions to promote well-being and prevent emotional distress [45].

Based on univariate analysis, inactive adolescents seemed to be more distressed compared to active adolescents. Moreover, those who had abandoned their commitment to a sports club during the pandemic showed greater distress than adolescents still engaged. However, this tendency was weak and not significantly evident in the regression model. Notably, in the free-text questions many adolescents expressed their difficulty in reconciling the demands of sports training with school and confinement. Therefore, in our sample, physical activity seems to have a multifaceted role in relation to well-being and distress.

In line with previous literature data, sleep also proved to be associated with distress, as the most distressed adolescents said they were sleeping less than in the pre-pandemic period, and that their sleep quality had worsened. Sleep difficulties emerged also in the free-text answers as an aspect of discomfort, together with excessive use of digital devices. Indeed, it is widely recognized that sleep problems and distress are associated with smartphone overuse, as the association between the widespread phenomenon of using digital devices at night and psychological distress was identified even before the pandemic [36].

Our study further shows that about 50% of the sample was at risk for potential smartphone overuse and that the SAS-SV score proved to be associated with the presence of distress, even though the association was weak [20]. What emerged in particular was the association between distress and overuse of digital devices when studying (therefore linked to distance learning during the pandemic), or out of boredom. This result seems to suggest that the physical and social restrictions imposed during the pandemic could have exacerbated pre-existing negative lifestyles and habits, such as overusing smartphones to relieve boredom. Interestingly, in a more in-depth analysis of the factors constituting smartphone overuse, it has been noted that “functional impairment”, i.e., the interference that adolescents feel due to smartphone overuse when carrying out daily tasks, was the most significant factor. This testifies to how an excessive use of the smartphone may lead to the perception of a fragmentation of attention and to increased difficulty in completing tasks, contributing to a further increase in distress. Furthermore, the use of digital devices was not associated with distress when used for leisure. As we know, during the pandemic the use of technology was very important for maintaining social contacts, despite physical distancing. Thus, this result seems to confirm what was found in a recent study, which demonstrated how connectedness, enabled by new virtual technologies, was a lifeline for young people, a buffer against the devastating effects of physical distancing [46].

Finally, the sheer percentages of adolescents who clearly stated that they had perceived an increase in their experiences of anxiety (63%), that they were afraid of getting sick (48%) and that they needed to resort more to psychological help (23%) further confirms the relevance of their perceived emotional distress. Moreover, the increase in access to emergency departments for acute psychiatric issues, especially in girls, found in many previous studies in different geographical areas towards the end of the pandemic, calls for some reflections regarding the long-term effects of what was experienced by the young population during the pandemic period [18,42,43,47,48].

Exploring the reasons that adolescents themselves identified as the cause of their distress opened a window into the deeper and inner world of young people’s emotional experiences. It is recognized that disclosing one’s emotions or difficulties is a truly difficult task for adolescents. Nevertheless, it seemed that many adolescents welcomed the opportunity to communicate aspects of their emotional experience, suggesting an underlying need to feel one’s discomfort listened to, perhaps favored by anonymity: more than 70% of young people with distress agreed to describe the reasons for their feelings, and those who perceived it most did so to a greater extent. Indeed, adolescents who exceeded the clinical threshold for distress were more likely to report profound psychological distress or relational problems in their free-text answers.

Adolescents’ answers showed that school was one major cause for distress, especially because of the pressure they felt to perform well in tests, to keep up with classes and commitments, in order to meet their parents’ and teachers’ expectations. Difficulties in interpersonal relationships, either with their families or their peer groups, were the second most frequently mentioned cause for distress; and, importantly, the third were profoundly negative emotional states such as anxiety, fear of failure, or lack of motivation, which in a worrying number of answers led to extreme behaviors such as eating disorders, self-harm, and suicidal ideation. The pandemic then appears as the fourth reason for distress, especially in terms of the deprivation of social contacts with peers and constraint in family ties; exactly the opposite of the natural growth trend of adolescence. Another worrying aspect is the fifth reason for distress, which highlights a difficulty in defining the inner causes of the discomfort, hinting at a confusion that adolescents could not define and therefore was even more difficult to master. Finally, expected aspects such as dissatisfaction related to physical appearance are typical themes of adolescence, while sleep problems, issues with food and the overuse of digital devices further confirmed the adolescents’ difficulties which emerged in the first part of the questionnaire.

The themes arising from the analysis of the free-text answers in our study suggest several reflections. First, we could find remarkable similarities with the results of a Korean study carried out before the pandemic, on 291,110 adolescents aged 12–18 years, where school-related tensions were the most frequently reported reason for distress, followed by conflicts with peers and difficulties in family relationships [49]. Also, in our sample, school-related tensions were the first reason for distress, followed by relationships with peers and family, demonstrating that these are fundamental issues for adolescents in our society, beyond the pandemic. Moreover, the authors of the Korean study found that relational difficulties were linked to more serious consequences for psychological well-being, such as depression and suicidal risk. In our study, too, we could notice the emergence of a widespread psychological malaise which seems to be particularly associated with relational difficulties. This emerges both by reading adolescents’ own words, and from the datum highlighting the fact that distress was significantly higher for those who reported relational difficulties as a reason for their distress (90% vs. 78%).

The intensity of the experiences of discomfort and the suffering expressed in some free-text answers of the adolescents in our study, certainly exacerbated by the pandemic condition, are particularly touching and make us question ourselves regarding possible interventions.

Stress resulting from school pressure is an aspect that has been particularly studied in recent years as a determinant of mental health problems [50]. An association between school pressure and depression, anxiety, self-harm, and suicide in the adolescent population from different countries has been recently documented in a meta-analysis study [51]. A further study highlighted the relationship between teachers’ caring behavior and the improvement in students’ socio-emotional competence and involvement in studying [52]. In our study, school pressure was related to the sense of being overwhelmed, the fear of failing or disregarding the expectations of significant adults, a lack of understanding from teachers, and the idea that psychological distress is not seen by adults, all aspects that can fuel distress and seriously threaten adolescent psychological well-being. Also the description of relationships experienced as toxic, rejecting, devaluing or extremely demanding, and the sense of loneliness and relief found in self-harm or suicidal thoughts in such young children, are particularly worrying aspects. We have not directly investigated distress in parenting due to the pandemic, but, in line with this result, other studies have highlighted the negative effects of parental distress on adolescents’ well-being [53,54].

In recent years, given the increase in youth problems at school, interventions that used the school environment as a privileged place to promote psychological well-being have multiplied and have proven effective [55,56]. School is the environment in which young people spend most of their time; the experiences they have there are formative and accompany them in their growth and formation of adult identity. School-based public health interventions on the prevention of emotional distress in schools have the advantage of intercepting all students and providing differentiated help, according to each individual’s needs. Of particular interest are universal interventions, such as the HORS-PISTE universal anxiety-prevention program, which are not only aimed at students experiencing full-blown symptoms but are aimed at promoting the psychosocial skills of the entire school community, with encouraging results [57]. The results of our study suggest looking at this type of intervention to promote well-being and actively prevent distress in adolescent students.

Some limitations of the current study should be underlined. First, the cross-sectional design did not allow for the establishment of causal relationships between the study variables and the presence of possible psychological distress. Second, the generalization of the results to Italian adolescents was limited, considering the low response rate of students and the fact that the survey was carried out in one of the cities in the North of Italy with the highest number of days of school closure. Third, the use of self-reported data might possibly bear some inaccuracies because of recall bias related to changes in lifestyle habits compared to the pre-pandemic period. Moreover, due to the criterion of free participation in the study for both schools and students, we cannot exclude a selection bias (probably, participants more sensitive to the theme were more willing to complete the survey). Finally, parental distress and parenting quality were not measured. Nevertheless, the results of this study are consistent with the evidence retrieved so far concerning the impact of the pandemic on the mental health of young people [27,58].

As the problems highlighted by the study, particularly distress related to school pressure, are emerging as increasingly widespread public health problems, future studies should address ED in Italian adolescents nationwide. It would also be important to evaluate the strengths and weaknesses of interventions aimed at adolescent well-being in the school setting, which seems to be the most suitable place for promoting psychological well-being of young people.

One of the major strengths of the study is the sample size, which was one of the largest among Italian studies on adolescents. In addition to this, the results were supported by many free open-ended questions that allow students to better reveal their vulnerabilities and allow us to better analyze the reasons for psychological distress. Finally, the survey explored several aspects of daily life including eating habits, physical and screen activity, sleep, and smartphone addiction, in relation to psychological distress, providing a unique contribution to the knowledge of the long-term impact of pandemic restrictions among adolescents.

## 5. Conclusions

The results of this study highlighted an increase in perceived distress among high school adolescents in the final part of the COVID-19 pandemic. This is an expected result, which underlines the effects of the restrictions and suffering linked to the pandemic but which brings to attention an already known problem, namely the increase in psychological suffering expressed by adolescents in the last decade. This phenomenon has recently been defined as a new public health problem that must be addressed with interventions that involve not only adolescents but also their teachers, families, the community and social policies.

Our study aimed to reveal the reasons that adolescents perceive as underlying their difficulties, to better understand how possible interventions can help them. The influence of the pandemic was found to be mainly linked to suffering due to physical distancing from peers and also to difficulties related to online teaching. This must be carefully taken into account in the event of new pandemic events.

On the other hand, the experiences of emotional distress expressed in the free-text answers have shed light on specific and profound aspects of adolescents’ distress, linked in particular to the school pressure and the quality of relationships with significant others such as peers, parents and teachers. We believe that these issues should be kept in mind when designing support interventions for adolescents.

Furthermore, the results of our study have highlighted the co-presence of disadvantageous lifestyles such as little sleep, poor nutrition, poor physical activity and prolonged use of screens associated with an increase in distress in approximately half of the adolescents examined two years after the onset of the COVID-19 pandemic. To avoid the long-term consequences of these unhealthy lifestyles, it is important to devise prevention programs which also take these aspects into account.

The impact of the pandemic has certainly been enormous; the recovery from all this seems to be more complex than it was possible to imagine, and we believe that what emerges from this study can help design interventions, taking into account the multidimensional needs of adolescents, to promote their global well-being.

## Figures and Tables

**Figure 1 behavsci-14-00775-f001:**
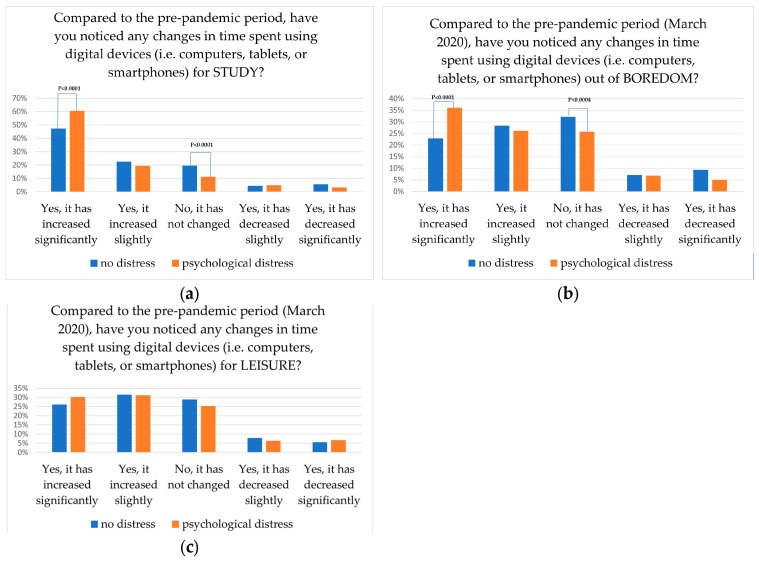
Distribution of responses related to time spent on device for study (**a**), leisure (**b**) and out of boredom (**c**), according to possible presence of psychological distress.

**Figure 2 behavsci-14-00775-f002:**
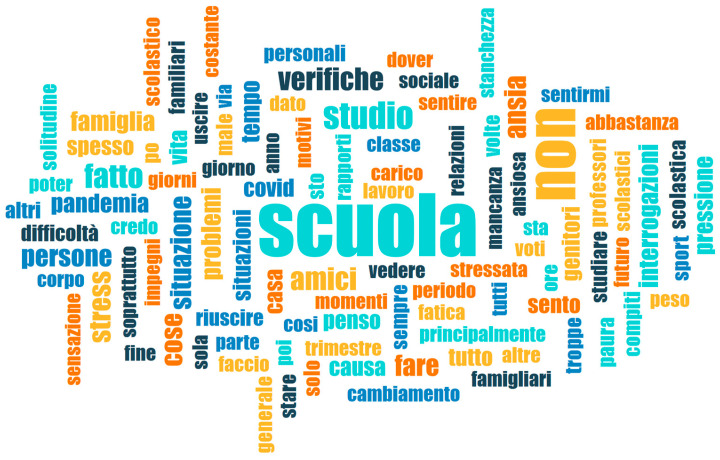
Word cloud showing the most frequent content words in the corpus of free-text answers, obtained using MaxQDA (Verbi Software 2024).

**Table 1 behavsci-14-00775-t001:** Univariate and multivariate analyses between predictors and possible presence of psychological distress.

	Presence of Distress	*p* Value	AdjOR (95% CI)	*p* Value
Variable	Univariate Analysis		Multivariate Analysis	
Age (Years)	N (%)	<0.0001		
13–15	369 (44.8)		Reference	
16–18	455 (56.2)		1.64 (1.29–2.08)	<0.0001
19–21	24 (45.3)		1.04 (0.53–2.04)	ns
Gender		<0.0001		
Male	251 (31.9)		Reference	
Female	567 (67.0)		2.27 (1.77–2.90)	<0.0001
Not specified	30 (55.6)		1.34 (0.68–2.63)	ns
Type of School		0.019		
State-funded	632 (48.7)		Reference	
Private	216 (55.5)		1.15 (0.79–1.66)	ns
Parents’ education (n = 1470)		0.053		
High school degree or less	427 (48.8)		Reference	
University degree ^1^	321 (54.0)		1.17 (0.86–1.58)	ns
Change in eating habits compared to pre-pandemic period (food amount)		<0.0001		
I eat as before	200 (35.1)		Reference	
I eat more	224 (51.5)		1.09 (0.79–1.51)	ns
I eat less	272 (69.7)		1.90 (1.35–2.69)	<0.0001)
I don’t know	152 (52.2)		1.24 (0.88–1.76)	ns
Change in eating habits compared to pre-pandemic period (food quality)		<0.0001		
They did not change	218 (33.9)		Reference	
They are healthier	219 (50.6)		1.21 (0.89–1.65)	ns
They are less healthy	129 (66.2)		1.63 (1.07–2.49)	0.024
They changed but I don’t know how	282 (68.1)		2.00 (1.45–2.78)	<0.0001
Weight change compared to pre-pandemic period		<0.0001		
My weight is the same	270 (42.7)		Reference	
My weight has increased	227 (49.2)		1.00 (0.69–1.47)	ns
My weight has decreased	236 (65.0)		1.49 (0.97–2.29)	ns
I don’t know/I prefer not to answer	115 (50.2)		0.91 (0.54–1.53)	ns
Comfort food consumption		<0.0001		
No/I don’t know	399 (37.9)		Reference	
Yes, more than before the pandemic	203 (73.0)		1.50 (1.05–2.14)	0.025
Yes, less than before the pandemic	65 (70.7)		1.88 (1.11–3.18)	0.019
Yes, both before and during the pandemic	181 (68.6)		2.02 (1.44–2.83)	<0.0001
Change in moderate physical activity compared to pre-pandemic period		<0.0001		
Unchanged	339 (45.4)		Reference	
Decreased	201 (61.5)		1.44 (0.98–2.11)	ns
Increased	308 (50.2)		1.17 (0.86–1.60)	ns
Physical activity ^2^		0.05		
Active	226 (46.6)		Reference	
Inactive	622 (51.8)		0.89 (0.64–1.24)	ns
Engagement in a sport club before the pandemic ^3^		0.026		
Yes No	554 (48.4) 294 (54.2)			
Maintenance of commitment with the sport club ^4^		0.049		
Yes No	290 (45.8) 264 (51.7)		Reference 0.78 (0.59–1.04)	ns
Sleep duration		<0.0001		
>8 h	254 (40.4)		Reference	
<8 h	594 (56.2)		1.45 (1.14–1.85)	0.003
Change in sleep quality compared to pre-pandemic period		<0.0001		
Unchanged	369 (40.6)		Reference	
Improved	135 (46.6)		0.89 (0.60–1.33)	ns
Worsened	344 (70.6)		1.23 (0.84–1.79)	ns
Smartphone addiction (mean ± SD)	33.7 ± 9.1 vs. 29.9 ± 8.9 ^5^	<0.0001	1.02 (1.0–1.03)	0.007
Increased anxiety perception compared to pre-pandemic period		<0.0001		
No/I don’t know/I prefer not to answer Yes	284 (30.4) 564 (74.8)		Reference 3.37 (2.88–4.68)	<0.0001
Increased fear of getting sick compared to pre-pandemic period		0.001		
No/I don’t know/I prefer not to answer	216 (58.2)		Reference	
Yes	632 (48.1)		0.95 (0.66–1.37)	ns
Increased need of health professional support compared to pre-pandemic period		<0.0001		
No/I prefer not to answer	649 (45.5)		Reference	
Yes	199 (76.8)		1.84 (1.29–2.62)	0.001

^1^ At least one parent with a university degree. ^2^ The physical-activity variable was created according to WHO recommendations. Adolescents were considered active if they declared doing at least an average of 60 min per day of moderate-to-vigorous intensity and vigorous-intensity aerobic activities, at least 3 days a week. ^3^ Variable not included in the logistic model because of collinearity. ^4^ Adolescents who declared being involved in a sport club before the pandemic answered this question (n = 1144). ^5^ Comparison with students without psychological distress.

## Data Availability

Data are available upon reasonable request to the authors.

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
