# Peer review of "How Distressed Are Adolescent Students? A Mix-Method Study on High School Students in Northern Italy, Two Years after the Beginning of the COVID-19 Pandemic"

_behavsci, 2024, doi:10.3390/bs14090775_

Round 1

Reviewer 1 Report

Comments and Suggestions for Authors

The abstract has a clear, functional and adequate structure. It provides information about its purpose, method, results and discussion. Its organization and writing are optimal.    

The introduction and theoretical contextualization of the manuscript is adequate, functional and current. It provides an adequate characterization regarding the main categories of analysis addressed in the manuscript, such as adolescence, the pandemic caused by the Sars-CoV-2 virus, stress, mental health and mental health disorders related to anxiety and eating behaviors. It provides a sufficient, pertinent and diverse context about the field in which the research is framed.    

The description of the method is adequate. It provides information about the study design, data collection techniques (instruments) and data analysis strategy. However, as a suggestion, I recommend structurally improving the description of the sample, using a table to describe the inclusion and exclusion criteria of participants, along with explaining the procedure for contacting and selecting schools belonging to the Brescia district. On the other hand, I would recommend explaining the ethical criteria that guided the study and whether or not it had the approval of a scientific ethics committee for work with human beings.    

The description of the results is adequate. They use tables, graphs and representative quotes to multidimensionally represent the most distinctive findings of the study. However, the qualitative description of the results is confusing and there is no logical or coherent sense in the wording. I recommend reviewing and structurally complementing this subsection.    

The discussion of the findings is diverse, multidimensional and includes an updated review of the state of the art on the variables analyzed in the study. In particular, there are valuable areas of analysis related to adolescent mental health in the context of the pandemic with its various implications on psychological well-being, eating patterns, sleep patterns, smartphone use, use of free time, physical activity, among others. They also establish interesting relationships about the impact of the pandemic on the presence of symptoms related to stress or anxiety. As a suggestion, I would recommend explaining the limitations and projections of the study from a theoretical and methodological point of view.

The conclusions are brief and generic. As a suggestion, an exhaustive review of the references would be desirable, in order to strictly comply with the editorial standards of the journal.

Author Response

Dear Reviewer

We would like to thank you for taking the time to carefully read and review our paper, and for your useful suggestions. We have modified the article following your suggestions and feel that this has significantly improved the quality of the paper. In a few cases, we could not comply with all your requests, but have tried to provide explanations for our choices.

Following is a point-by-point response to each comment.

The abstract has a clear, functional and adequate structure. It provides information about its purpose, method, results and discussion. Its organization and writing are optimal.    

The introduction and theoretical contextualization of the manuscript is adequate, functional and current. It provides an adequate characterization regarding the main categories of analysis addressed in the manuscript, such as adolescence, the pandemic caused by the Sars-CoV-2 virus, stress, mental health and mental health disorders related to anxiety and eating behaviors. It provides a sufficient, pertinent and diverse context about the field in which the research is framed.    

The description of the method is adequate. It provides information about the study design, data collection techniques (instruments) and data analysis strategy. However, as a suggestion, I recommend structurally improving the description of the sample, using a table to describe the inclusion and exclusion criteria of participants, along with explaining the procedure for contacting and selecting schools belonging to the Brescia district. On the other hand, I would recommend explaining the ethical criteria that guided the study and whether or not it had the approval of a scientific ethics committee for work with human beings.

As requested, we have better explained (in new line 172) that all students attending involved schools and willing to participate, could access the online survey; no further selection criteria was required. We contacted all high schools in Brescia District; among them eight schools voluntarily agreed to participate in the study (new lines 173-176); within each involved schools, every student was free to decide if complete the survey or not. We added the reference to Italian Law about ethical issues of the study, and we better specify that further methodological details of the study have been published elsewhere (new lines 176-179 and 183-185).

The description of the results is adequate. They use tables, graphs and representative quotes to multidimensionally represent the most distinctive findings of the study. However, the qualitative description of the results is confusing and there is no logical or coherent sense in the wording. I recommend reviewing and structurally complementing this subsection.    

Thank you for this suggestion. We have focused on the subsection describing the results of the content analysis of the free-text answers. In order to make the results section clearer, we first decided to describe more clearly how the analysis was carried out in the methods section (lines 263-273 and 341). Then, we rewrote the first paragraph of section 3.1.2. hoping to better introduce the thematic analysis of the free-text answers (lines 373-416). We also modified the ending and introductory paragraphs of sections 3.1.2. and 3.1.3 (lines 373-382, 414-416 and 418-420) hoping to better explain the logical flow of the section.

The discussion of the findings is diverse, multidimensional and includes an updated review of the state of the art on the variables analyzed in the study. In particular, there are valuable areas of analysis related to adolescent mental health in the context of the pandemic with its various implications on psychological well-being, eating patterns, sleep patterns, smartphone use, use of free time, physical activity, among others. They also establish interesting relationships about the impact of the pandemic on the presence of symptoms related to stress or anxiety. As a suggestion, I would recommend explaining the limitations and projections of the study from a theoretical and methodological point of view.

Thanks for the comment. Several limitations of the study have already been reported (lines 706-718), but according to this suggestion, we agreed to add a further methodological limitation, related to a possible selection bias (new lines 713-715). The projections have also been included (lines 719-724) and expanded as reported below:

As the problems highlighted by the study, particularly distress related to school pressure, are emerging as increasingly widespread public health problems, future studies should address ED in Italian adolescents nationwide. It would also be important to evaluate the strengths and weaknesses of interventions aimed at adolescent well-being in the school setting which seems to be the most suitable place for promoting psychological well-being of young people.

The conclusions are brief and generic.

Thanks for the comment, the conclusions have been expanded (lines 734-761) as reported below:

The results of this study highlighted an increase in perceived distress among high school adolescents in the final part of the Covid-19 pandemic. This is an expected result which underlines the effects of the restrictions and suffering linked to the pandemic but which brings to attention an already known problem, namely the increase in psychological suffering expressed by adolescents in the last decade. This phenomenon has recently been defined as a new public health problem that must be addressed with interventions that involve not only adolescents but also their teachers, families, the community and social policies.

 Our study aimed to reveal the reasons that adolescents perceive as underlying their difficulties to better understand how possible interventions can help them. The influence of the pandemic was found to be mainly linked to suffering due to physical distancing from peers and also to difficulties related to online teaching. This must be carefully taken into account in the event of new pandemic events.

On the other hand, the experiences of emotional distress expressed in the free-text answers have shed light on specific and profound aspects of adolescents’ distress, linked in particular to the school pressure and the quality of relationships with significant others such as peers, parents and teachers. We believe that these issues should be kept in mind when designing support interventions for adolescents.

Furthermore, the results of our study have highlighted the co-presence of disadvantageous lifestyles such as little sleep, poor nutrition, poor physical activity and prolonged use of screens associated with an increase in distress in approximately half of the adolescents examined two years after the onset of the Covid-19 pandemic. To avoid the long-term consequences of these unhealthy lifestyles, it is important to devise prevention programs taking also these aspects into account.

 The impact of the pandemic has certainly been enormous, the recovery from all this seems to be more complex than it was possible to imagine, and we believe that what emerges from this study can help design interventions taking into account the multidimensional needs of adolescents to promote their global well-being.

As a suggestion, an exhaustive review of the references would be desirable, in order to strictly comply with the editorial standards of the journal.

Thank you for the through reading; according to the journal's instructions, the references were listed using Zotero software, ACS style. Probably, the version used generated parentheses instead of full stops in the bibliography; the difference has been resolved for all citations. Moreover, the citations placed in the middle of the sentence have been moved before the punctuation, according to editorial standards (lines 129, 555, 558, 620 and 660).

Reviewer 2 Report

Comments and Suggestions for Authors

Comments to the authors

The manuscript is clear, and its aims, which have the potential to significantly advance the existing literature in the field, are relevant.

Abstract

- At some point, the authors mentioned the statistical values, but to be honest, I’m not used to seeing the P values in the abstract.

- It could be interesting to present the year's range of adolescents in the study, as adolescence covers a wide age range, and this information could be useful for the readers.

- “associated to ED were female gender.” I will suggest presenting this result differently; in fact, gender is the variable/factor, with females presenting more ED than boys, right?

- While the focus on ED is clear, the conclusions could benefit from a more comprehensive exploration of the other variables, such as anxiety perception, age, and sleep. This would provide a complete picture of the research and its potential impact on academic and practical contexts. 

Introduction

- The introduction is very well written. I will probably focus more on the variables studied, with a short conceptual framework. The authors reinforce the impact of COVID-19 around the world and on adolescents. However, concepts such as quality of life and emotional distress are not presented explicitly. In my opinion, readers will benefit from these “definitions”.

- After presenting all the results of previous literature, at the end of the introduction, what are the main hypotheses?

Results 

- The quality of images could be improved (p.8).

- Honestly, I would prefer to see the data with the frequency of words than the word-cloud.

- I think the information presented with bullets (line 355-XX/line 379-XX) could be in text form.

Line 385, the term “accuse” could be very strong for the context. The terms report/mention...could be more appropriate.

Discussion

Line 511: the explanation for the present result was the neurodevelopment of adolescents in these phases and it seems a good explanation. However, due to the comments about the school, considering that these students will move to universities, it could not be this change to other contexts, and even the pressure to enter a specific university or course also influences these results. As you mentioned, schools are one of the principal factors for distress.

Author Response

Dear Reviewer

We would like to thank you for taking the time to carefully read and review our paper, and for your useful suggestions. We have modified the article following your suggestions and feel that this has significantly improved the quality of the paper. In a few cases, we could not comply with all your requests, but have tried to provide explanations for our choices.

Following is a point-by-point response to each comment.

Comments to the authors

The manuscript is clear, and its aims, which have the potential to significantly advance the existing literature in the field, are relevant.

Abstract

- At some point, the authors mentioned the statistical values, but to be honest, I’m not used to seeing the P values in the abstract.

Thank you for this suggestion; we have deleted references to the p value from the abstract.

- It could be interesting to present the year's range of adolescents in the study, as adolescence covers a wide age range, and this information could be useful for the readers.

Thank you for this suggestion; we have added this information to the abstract (lines 15-16).

- “associated to ED were female gender.” I will suggest presenting this result differently; in fact, gender is the variable/factor, with females presenting more ED than boys, right?

Thank you for this remark; we have reformulated the sentence accordingly (lines 19-20).

- While the focus on ED is clear, the conclusions could benefit from a more comprehensive exploration of the other variables, such as anxiety perception, age, and sleep. This would provide a complete picture of the research and its potential impact on academic and practical contexts. 

Thank you for your thorough remarks concerning the abstract. We have revised the text according to your previous suggestions and think that it is now clearer in its exposition of the results (lines 19-23). Unfortunately, we could not further expand the conclusion section of the abstract for reasons of space (200 words max).

Introduction

- The introduction is very well written. I will probably focus more on the variables studied, with a short conceptual framework. The authors reinforce the impact of COVID-19 around the world and on adolescents. However, concepts such as quality of life and emotional distress are not presented explicitly. In my opinion, readers will benefit from these “definitions”.

Thank you for this suggestion; we have added the following definitions (lines 102-105 and 141-146):

“Quality of life”: defined as a broad concept typically referring to an individual’s overall well-being, encompassing physical health, psychological state, level of independence, social relationships, personal beliefs, and their relationship to salient features of the environment (WHO 1997).

Reference: World Health Organization. (1997). WHOQOL: Measuring Quality of Life. Division of Mental Health and Prevention of Substance Abuse, World Health Organization. Available from: [https://apps.who.int/iris/handle/10665/63482](https://apps.who.int/iris/handle/10665/63482).

“Emotional distress”: The presence of emotional distress in adolescents during the pandemic has been documented in numerous studies (Jost et al. 2023) as an increase in psychological symptoms including anxiety, sadness, depression, loneliness, restlessness and fear. Specific vulnerabilities of adolescence are also the fear of failure, self-doubt and loss of motivation [ Erikson, 1968, 12]. Themes that we expect to emerge in free-text answers.

Reference: Jost, G. M., Hang, S., Shaikh, U., & Hostinar, C. E. (2023). Understanding adolescent stress during the COVID-19 pandemic. Current Opinion in Psychology52, 101646.

- After presenting all the results of previous literature, at the end of the introduction, what are the main hypotheses?

Thank you for pointing out that the hypotheses were not expressed clearly in the introduction. We have added the following paragraph (lines 154-162):

Considering that such a large study on the Italian adolescent population has not been carried out before, but in line with previous research carried out in other geographical contexts, we hypothesized that also in our sample emotional distress would increase compared to the pre-pandemic period. We also hypothesized that ED would be associated with different aspects of lifestyle such as eating habits, difficulty sleeping, overuse of smartphones and digital devices. Finally, we hypothesized that ED would be associated with the presence of difficulties related to school commitments and to relationships with peers and with relatives, possibly exacerbated by the restrictions and changes imposed by the pandemic.

Results 

- The quality of images could be improved (p.8).

Thank you for pointing this out. We have already provided the editors with a separate file of the images in the requested format; we have used the template provided by the journal to embed the images in the text. We hope that they will be more easily readable in the final formatted version of the manuscript.

- Honestly, I would prefer to see the data with the frequency of words than the word-cloud.

We have carefully considered your suggestion, and added the frequencies of the words mentioned in the text accordingly (lines 354-364). However, we would like to retain the WordCloud in the main text, and to keep the full list of the most frequent words in the corpus, their frequencies and their translation into English, in the Supplementary Materials. We made this decision because the Table is very lengthy and fear that it incorporating it into the text would disrupt the flow of the reading.

We would also like to emphasise that, as specified in the Methods section (lines 253-259), the size of each word in the WordCloud represents that word’s frequency in the corpus; this is why we feel that the image describes the frequencies of the words in the corpus with more immediacy.  

- I think the information presented with bullets (line 355-XX/line 379-XX) could be in text form.

Thank you for this suggestion; we have modified the text accordingly.

Line 385, the term “accuse” could be very strong for the context. The terms report/mention...could be more appropriate.

Thank you; we have substituted “accuse” with “report”.

Discussion

Line 511: the explanation for the present result was the neurodevelopment of adolescents in these phases and it seems a good explanation. However, due to the comments about the school, considering that these students will move to universities, it could not be this change to other contexts, and even the pressure to enter a specific university or course also influences these results. As you mentioned, schools are one of the principal factors for distress.

Thank you for this useful suggestion. We have added the following paragraph (lines 569-574):

Furthermore, the phase between 16 and 18 years is the one in which young people are most focused on choices related to their academic future. These are influenced by the skills they feel they have acquired and therefore the sensitivity to school failure is very high in this phase. Therefore, the school pressure linked to the changes due to the pandemic, as highlighted by the answers to the free-text questions, may also be at the root of the increase in ED.